# Preparation and Lubricating Properties of Polystyrene Composite Microspheres

**DOI:** 10.3390/ma16083071

**Published:** 2023-04-13

**Authors:** Wen Zeng, Weiqing Huang, Bing Guo, Yang Sun, Hangyan Shen

**Affiliations:** 1College of Materials and Chemistry, China Jiliang University, Hangzhou 310018, China; 2Hangzhou Huaguang Advanced Welding Materials Co., Hangzhou 311112, China

**Keywords:** composite microspheres, lubricating properties, polystyrene, PTFE

## Abstract

In order to improve the lubrication performance of polystyrene microspheres (PS) as solid lubricant in drilling fluids, elastic graphite–polystyrene composite microspheres (EGR/PS), montmorillonite–elastic graphite–polystyrene composite microspheres (OMMT/EGR/PS), and polytetrafluoroethylene–polystyrene composite microspheres (PTFE/PS) were prepared by suspension polymerization. OMMT/EGR/PS has a rough surface, while the surfaces of the other three composite microspheres are smooth. Among the four kinds of composite microspheres, the largest particle is OMMT/EGR/PS, and the average size is about 400 μm. The smallest particle is PTFE/PS, and the average size is about 49 μm. Compared with pure water, the friction coefficient of PS, EGR/PS, OMMT/EGR/PS and PTFE/PS reduced by 25%, 28%, 48%, and 62%, respectively. The wear tracks of EGR/PS, OMMT/EGR/PS and PTFE/PS are narrower and smoother than those of pure water. When the content of PTFE is 4.0 wt%, the friction coefficient and wear volume of PTFE/PS are 0.213 and 2.45 × 10^−4^ mm^3^—74% and 92.4% lower than that of pure water, respectively.

## 1. Introduction

High-performance lubricants are important means to reduce friction and wear of mechanical equipment, improve operation efficiency and service life, and reduce energy consumption. With the rapid development of industry, lubricants based on mineral oil and synthetic oil have caused harm to the environment, and substances in mineral oil have directly threatened human health. Water-based green lubricants are environmentally friendly. Green lubricants are also gradually being prepared by scholars [1]. It is an important direction for lubricating materials and lubricating additives in the future, especially for oil exploitation fields [2,3].

With the rapid development of oil exploration work, the drilling depth is increasing, and various stuck-pipe accidents have occurred frequently. Decreasing the torque of drilling tools and reducing the wear and tear of drilling tools to avoid sticking accidents have become problems that need to be solved urgently. Solid lubricating materials were used to prevent and protect the friction surface from damage during relative movement and reduce its friction and wear [4]. Solid lubricants are widely used in mobile machinery components and are very important to the development of mobile machinery. For example, the use of liquid lubrication is limited if operating conditions are outside the fluid domain, such as high temperatures, or if liquid cannot be introduced [5].

Adding high-efficiency solid lubricants to drilling fluid is one of the effective methods of decreasing drilling tool torque, reducing drilling tool wear and avoiding stuck-pipe accidents. As a common solid lubricant for drilling fluids, polystyrene microspheres can convert surface contact friction into point contact friction and effectively reduce the frictional resistance of drilling tools [6].

Compared with other solid lubricants, polymer materials such as polystyrene microspheres (PS) have significant advantages, such as good toughness, not damaging the dual material, good chemical stability and not reacting with other materials [7,8]. However, PS bearing capacity and high-temperature resistance are poor, with easily softened deformation in the use process, thus affecting the lubrication effect. One of the effective ways to solve the above problems is preparing composite microspheres by polymerizing styrene and other materials.

Montmorillonite (MMT) is a natural layered silicate commonly used in polymer modification, usually for the preparation of polymer–clay nanocomposites, because MMT can improve the thermal stability and mechanical properties of polymers. As such, the composite material has good heat-resistance stability and high strength and stiffness [9,10]. It can also improve the load-bearing capacity of the composite microspheres. Elastic graphite (EGR) has the advantages of high purity, good flexibility, and high elasticity. It has stable performance and temperature resistance [11], and does not react chemically with other drilling fluid additives. EGR is also a good lubricant material [12].

Polytetrafluoroethylene (PTFE) is a high-molecular-weight polymer obtained by polymerizing tetrafluoroethylene monomer. It has high-temperature resistance characteristics, so it can improve composite microspheres’ high-temperature resistance. It has a very low friction coefficient, so it can be used as an additive with excellent self-lubricating properties and has been widely used in solid lubricating materials and greases [13,14]. It is worth noting that substances like PTFE decompose extremely slowly under ambient conditions, so they will accumulate in organisms and the environment and cause harm [15], but considering the complete alternative will entail huge cost, we can ameliorate these problems through its production process [16] and recycling research. At the same time, the use of PTFE’s excellent self-lubricating properties can also be maximized [17]. Therefore, PTFE still has important application value in lubricant additives.

In this study, to further improve the lubricating performances of PS, organic montmorillonite (OMMT), elastic graphite (EGR) and PTFE were used as additives to be compounded with styrene by suspension polymerization methods to prepare three kinds of polystyrene composite microspheres. The composite microspheres were successfully tested as additives for high-performance lubricants [18]. The effects of additive components on the tribological properties of polystyrene microsphere lubricants were investigated, and the composite microspheres’ lubrication mechanism is discussed and revealed.

## 2. Materials and Methods

### 2.1. Materials

The organic modified OMMT was purchased from Zhejiang Feng Hong New Materials Factory. The surface topography of OMMT is shown in Figure 1a. OMMT has a lamellar structure, and the length of a single piece is more than 2 μm. The EGR (TXT-500) was purchased from Qingdao Yan Hai Carbon Materials Company. The surface topography of EGR is shown in Figure 1b. As can be seen from the Figure 1b, the surface of EGR is smooth. A single piece is about 1–2 μm in length and 0.5–1 μm in width. The PTFE dispersion was purchased from Macklin Biochemical Company in Shanghai, China. The surface topography of PTFE particles is shown in Figure 1c. PTFE is about 0.3–0.5 μm in length and approximately 0.2 μm in width. Polyvinyl alcohol (PVA) with a polymerization degree of 1700 and degree of hydrolysis of 88%, styrene (St), divinylbenzene (DVB) and benzoyl peroxide (BPO) were also purchased from Macklin Biochemical Company.

### 2.2. Preparation of Composite Microspheres

#### 2.2.1. Preparation of EGR/PS

Before preparing the microspheres, the aqueous phase solution was prepared as follows. PVA was dissolved in deionized water at a stirring speed of 100 rpm at room temperature, heated to 80 °C, and held for 4 h until wholly dissolved. After the reaction, the product was 2.5% PVA solution. Then, the PVA solution was diluted to the corresponding concentration according to the experimental needs. After that, the organic phase solution was prepared. Cross-linker DVB (2 g) and 0.36 g initiator BPO were added to 18 g St monomer solution. Then, 0.5 g EGR was dispersed in the mixed solution until mixed well. Then, aqueous phase and organic phase solutions were mixed in a volume ratio of 3:1. The polymerization proceeded under mechanical stirring of 500 rpm at 80 °C for 8 h. The polymerization was performed in a three-neck flask in air. The prepared products were collected by vacuum filtration, then washed with distilled water and ethanol successively several times to remove soluble impurities. After vacuum drying, EGR/PS composite microspheres were obtained.

#### 2.2.2. Preparation of OMMT/EGR/PS

Similar to the preparation process of the EGR/PS in Section 2.2.1, the only difference being 0.25 g EGR and 0.25 g OMMT added into the St mixed solution and sonicated until mixing evenly in the organic phase preparation.

#### 2.2.3. Preparation of PTFE/PS

The PTFE dispersion was added to the prepared PVA solution. The resulting mixed solution was used as an aqueous phase solution. The mass ratios of PTFE to St monomer were 0.3 wt%, 0.5 wt%, 1.0 wt%, 2.0 wt%, 3.0 wt% and 4.0 wt% respectively. Then, 18 g St, 2 g DVB and 0.36 g BPO were stirred until thoroughly combined as an organic phase, then a polymerization reaction was carried out. The polymerization conditions and subsequent treatment of the productions are the same as those described in Section 2.2.1.

#### 2.2.4. Preparation of PS

Similar to the preparation process of the EGR/PS in Section 2.2.1., the difference being the organic phase solution was only composed of St, BPO, and DVB, and no other materials were added.

### 2.3. Characterization

The morphology of polystyrene composite microspheres and wear tracks were characterized by field-emission scanning electron microscopy (JSM-7001F). Energy-dispersive X-ray spectroscopy (EDS) was used to analyze the composition of composite microspheres and wear tracks. The infrared absorption spectrum of polymer composites was measured by a Fourier-transform infrared spectrometer (Nexus470) to analyze the chemical structure. Cross sections of composite microspheres before and after treatment with acetone solution were observed using an inverted metallographic microscope (MDS400).

### 2.4. Friction Performance Test

The composite microspheres were uniformly dispersed in deionized water with the sonicator, and the mass ratio of composite microspheres to water was 4:100. During the friction test of each group, 50 mL composite microsphere lubricating dispersion was added to the friction tray. The friction and wear performance were tested using a reciprocating friction tester (HSR-2M). The tester reciprocated in a ball-on-disk contact type. A silicon nitride ball (Φ4 mm) was on the top, and a polished 304 stainless-steel sheet was used as the disk. The stainless-steel sheet reciprocated under a load of 10 N. The displacement amplitude was 20 mm, the reciprocating sliding frequency was 1 Hz, and the friction duration was 10 min. Each test was conducted three times, and the average value of three test results was taken as the test value. The step profiler (P-6) was used to observe the wear marks left by the reciprocating friction on the substrate. This test also was done three times, and the average value of the tests was taken as the test result.

## 3. Results and Discussion

### 3.1. Morphology Analysis

Figure 2 shows the microscopic morphologies of PS, EGR/PS, OMMT/EGR/PS and PTFE/PS. The composite microspheres of three kinds of composite materials have excellent spherical shape. It can be seen from Figure 2a,c,g that the composite microsphere shape is complete and evenly dispersed. Figure 2e shows that the OMMT/EGR/PS has a rougher surface than other composite microspheres and some holes can be observed. This may be due to the anti-polymerization effect of organic montmorillonite, which makes a small number of monomers and oligomers adsorb on the surface of organic montmorillonite. In the process of washing, the monomers and oligomers on the surface of microspheres were washed off by water and ethanol, thus forming many small pits and making the microspheres’ surface rough [19].

The particle size of composite microspheres in the SEM image was analyzed by software, and results are shown in Figure 2. At least 50 particles were used in the particle size distribution analysis for each sample. It can be seen from Figure 2b–f,h that the average particle sizes of composite microspheres are 62 μm, 150 μm, 400 μm and 49 μm, respectively. OMMT/EGR/PS is the largest of these composite microspheres and the smallest composite microsphere is PTFE/PS.

The reason for the larger particles of composite microspheres could be that large OMMT and EGR can play the role of skeleton in the droplets, which makes it difficult for droplets to be broken and they tend to agglomerate. Therefore, the OMMT/EGR/PS particles become larger than other kinds of composite microsphere. The average particle size of the PTFE/PS composite microspheres is slightly less than that of PS composite microspheres. Because the particle size of PTFE is only 0.2–0.4 μm, it will not affect the size of composite microspheres.

### 3.2. Infrared Analysis

Figure 3 shows the infrared spectra of three kinds of polystyrene composite microspheres. As shown in Figure 3a, the two absorption peaks occurring at 3057 cm^−1^ and 3026 cm^−1^ are generated by the stretching vibration of the C-H bond of the benzene ring. The deformational vibration of the benzene ring skeleton generates compact characteristic peaks from 1602 cm^−1^ to 1430 cm^−1^ [19]. The peaks at 756 cm^−1^ and 698 cm^−1^ are characteristic peaks formed by deformational vibrations in the benzene ring. The peaks at 1035 cm^−1^ correspond to the stretching vibration of the Si-O bond, and the peaks at 528 cm^−1^ correspond to the stretching vibration of the Al-O bond. They are the characteristic absorption peaks of OMMT. The absorption peak at 1630 cm^−1^ may be the bending vibration of C-OH, and the peak at 1070 cm^−1^ can be ascribed to the C-O-C bond, which is the characteristic peak of elastic graphite. It indicates that OMMT/EGR/PS is successfully prepared.

Similar to the characteristic absorption peak at 1070 cm^−1^ of elastic graphite, the peaks at 3057 cm^−1^–3026 cm^−1^, 1602 cm^−1^–1430 cm^−1^ and 756 cm^−1^–698 cm^−1^ of the benzene ring are shown in Figure 3b, indicating the EGR/PS is successfully prepared. It can be seen from Figure 3c, in addition to the characteristic peaks in the benzene ring mentioned above, that the peaks at 1215 cm^−1^ and 1155 cm^−1^ correspond to the antisymmetric and symmetric stretching vibration of CF_2_ groups in the PTFE chain. These characteristic peaks indicated that PTFE is dispersed in the polymer to form PTFE/PS.

### 3.3. Formation Mechanism of Composite Microspheres

#### 3.3.1. OMMT/EGR/PS

In order to understand composite microspheres’ internal structure and formation mechanism, the internal structure of composite microspheres was observed by inverted metallographic microscopy. The microspheres were inlaid in resin by an automatic mosaic machine, then the sample was ground and polished by a metallographic grinding machine and polisher. The surface of the microspheres was etched with acetone solution.

Figure 4a shows the SEM images of the OMMT/EGR/PS cross section. There are apparent lamellar structures in the microspheres, indicating that the polymerization process did not change the lamellar structure of OMMT or EGR, so the lamellar structure of OMMT and EGR is retained inside the microspheres. Figure 4b–f give the EDS result of composite microspheres. The uniform distribution of Si and Al elements in the microspheres proves the successful composite of OMMT and PS. The formation process of OMMT/EGR/PS is shown in Figure 4g. Because there is a strong van der Waals force between the lamellae of OMMT [20,21], the polymerization process did not destroy the lamellar structure of OMMT [22]. St monomers adsorbed on the surface of OMMT and EGR sheets, and composite microspheres were formed as the polymerization reaction proceeded.

#### 3.3.2. PTFE/PS

Figure 5a,b show the cross section of PTFE/PS before and after corrosion with acetone solution using a metallographic microscope. It can be seen that the surface of the microspheres is smooth in Figure 5a, but it can be seen in Figure 5b that the microspheres show a rough surface after acetone corrosion. Some white-powdered particles are dispersed over the surface, indicating that the PTFE particles may not be corroded.

SEM and EDS analyses confirm the hypotheses. Figure 5c gives the SEM image of samples after corrosion. EDS analysis was carried out after the amplification of the white powdered particles, and the results show the existence of elemental F. The presence of F proves that these white-powdered particles are PTFE not corroded by acetone. The formation process of the composite microspheres is shown in Figure 5d. The St monomer was dispersed into tiny droplets under stirring and the effect of dispersing agent. The PTFE particles were dispersed in the PVA dispersant solution. As the reaction progressed, they were enclosed in the St droplets or adhered on the surface of tiny droplets. These small droplets gradually aggregated into large droplets, forming PTFE/PS in which PTFE was dispersed.

### 3.4. Frictional Properties of Composite Microspheres

#### 3.4.1. Result of Friction and Wear Performance Test

Figure 6 shows the friction coefficient and wear rate of four kinds of microspheres and pure water under the same experimental conditions. Compared to pure water, the PS reduced the friction coefficient from 0.825 to 0.624. The friction coefficient reduced by about 25%, which indicates that the PS have lubricating properties. The lubricating properties of PS were further improved after adding different composite materials. Among them, the friction coefficient of EGR/PS and OMMT/EGR/PS reduced to 0.592 and 0.433, respectively, revealing a decrease of 28% and 48%. PTFE/PS shows the best antifriction effect, and the friction coefficient was reduced to 0.316. This is about a 62% reduction compared with pure water.

To investigate the anti-wear performance of composite microspheres, a wear analysis test was carried out on ear tracks on the stainless-steel substrate of the friction test. Figure 6b illustrates the wear volumes of the worn substrate under the same test conditions. It can be seen that adding composite microspheres can reduce the wear volumes of stainless substrate to different degrees. The wear volume of samples using PS is 27% lower than that of pure water. The PS compounded with different materials can further reduce the wear volumes. The wear volumes of the worn substrate using EGR/PS, OMMT/EGR/PS and PTFE/PS as lubrication additives are 38%, 58%, and 86% lower than that of pure water, respectively. PTFE/PS has the best anti-wear performance.

In order to explore the influence of the particle size of composite microspheres on their lubrication performance, as shown in Table 1, a series of PTFE/PS with different particle sizes were prepared by different polymerization conditions and friction test s carried out. The test results are shown in Table 1.

From Table 1, it can be seen that the friction coefficient of PTFE/PS gradually decreases and the lubrication performance gradually becomes better as the particle size of microspheres decreases. When the particle size of PTFE/PS containing 0.3 wt% PTFE decreased from 79.35 μm to 49.00 μm, the friction coefficient decreased from 0.412 to 0.372. The reduction of friction coefficient is due to the decrease in particle size of composite microspheres, which made it easier for the microspheres to enter between the friction surfaces and played the role of ball bearings, enhancing the effect of lubrication. When the PTFE content was increased from 0.3 wt% to 0.5 wt%, the PTFE/PS particle size changed little, but the friction coefficient decreased significantly. This increase in PTFE concentration enhanced the lubricating performance of PTFE/PS.

To further investigate the friction properties of PTFE/PS, Figure 7a shows the friction coefficients of PTFE/PS adding to different amounts of PTFE. As shown in the figure, the 4.0 wt% PTFE/PS sample has the best friction properties, and can reduce the friction coefficient to 0.213, lower by nearly 74% compared to pure water. The result indicates that increasing the PTFE content can improve the lubricating properties of PTFE/PS. Figure 7b shows the substrate’s wear volume of worn tracks using PTFE/PS. It can be seen that the wear volume of the PTFE/PS sample is much smaller than other samples, and with the increase in PTFE content in the composite microspheres, the wear volumes will accordingly decrease. The wear volume of PTFE/PS with 4.0 wt% PTFE was 92.4% lower than that of pure water, which indicates that the anti-wear performance of PTFE/PS is also enhanced with increased PTFE concentration.

#### 3.4.2. Analysis of Composite Microspheres’ Wear Tracks

The lubrication mechanism of composite microspheres was studied. Figure 8a–d show the wear tracks morphology of stainless-steel substrates with composite microspheres and pure water. It can be seen that the wear track of pure water is the widest and worst. Among the composite microspheres, the widest is the EGR/PS and the narrowest is the PTFE/PS. Figure 8e–h show the 2D scan image of each wear track. The 2D scan image of the wear tracks can correspond to the width of the wear tracks in the SEM image. The roughness of each wear track is shown in Figure 8i.

The roughness of wear track with pure water is about 197.00 nm, the width of the wear track is about 312.5 μm and the depth is 2.85 μm. This is a typical adhesive wear and furrow characteristic caused by silicon nitride balls. Large amounts of wear debris accumulated on both sides of the mark, forming abrasive wear and aggravating the wear volume.

Compared with the pure water, the wear tracks after adding composite microspheres are much smoother and the depth and width of wear tracks shallower. The worn surface shows only slight furrows, so the wear volumes decreased in varying degrees. The wear track width of EGR/PS is about 279.0 μm, the depth is about 2.03 μm, and the roughness is 175.88 nm. This shows that adding EGR/PS to water can improve the friction performance. However, the friction performance of EGR/PS is the worst among all the composite microspheres. The width of OMMT/EGR/PS is about 242.0 μm, the depth is about 1.53 μm, and the roughness is 150.40 nm. Among all the composite microspheres, PTFE/PS is the best. The roughness of the worn substrate of PTFE/PS with 4.0 wt% PTFE is only about 44.22 nm, and the width and depth of wear track is about 150.0 μm and 0.75 μm, much narrower and shallower than other wear tracks. The surface of the wear tracks is smooth, not even covering the polishing marks itself, proving that the PTFE/PS can play an obvious antifriction and anti-wear effect.

#### 3.4.3. The Lubrication Mechanism of PTFE/PS

The microspheres play a role as rolling bearings, which can change the sliding friction into rolling friction. It is one of the important reasons why composite microspheres can be used as a lubricant. To explore the reason why PTFE/PS has the best friction effect in all composite microspheres, the wear track of PTFE/PS was further analyzed.

Figure 9 shows the EDS analysis of wear track surface after sliding with PTFE/PS. Figure 9b,c show that there are two main elements, C and F, on the surface of wear tracks, and each element is uniformly distributed over the whole surface. Combined with the friction and wear test results (Figure 6 and Figure 7), it can be considered that PTFE/PS formed a protective transfer film on the substrate surface during the friction test [23]. Transfer film [24] effectively reduces the friction between the silicon nitride balls and stainless-steel substrate. As reported by relevant research, when polymer composites slide against metal counterparts [25], the transfer film has an essential influence on reducing friction and wear, playing a significant role in lubrication, so the PTFE/PS has excellent low friction coefficient and high wear resistance properties.

Based on the above results, the mechanism that governed the frictional behavior of PTFE/PS composite lubricating additive is processed, as depicted in Figure 10. During the friction sliding tests (Figure 10a), PTFE/PS can be applied as an excellent water-based lubricating additive. PTFE/PS changes the sliding friction into rolling friction while forming a transfer film. The formed PTFE transfer film protects composite microspheres from wear damage as much as possible, thus maximizing the lubrication and anti-wear performances.

## 4. Conclusions

Three kinds of polystyrene (PS) composite microspheres were synthesized by suspension polymerization, which were EGR/PS, OMMT/EGR/PS, and PTFE/PS. The surface morphology, chemical composition, particle size and tribological properties of these composite microspheres were carefully studied. The conclusions are as follows.

EGR/PS, OMMT/EGR/PS, and PTFE/PS were successfully prepared. The particle size of various composite microspheres was counted and compared, and the particles of PTFE/PS were the smallest, averaging 49 μm. The experimental operation of composite microspheres prepared by suspension polymerization was relatively simple. However, the particle size distribution of the prepared microspheres was very broad, and it was not easy to control the particle size of the microspheres in a very accurate range.Compared with pure water, PS, OMMT/EGR/PS, EGR/PS and PTFE/PS all improved the lubrication and wear resistance. PTFE/PS has the best lubricating and anti-wear properties, which can effectively reduce the friction coefficient and reduce the wear volume. Under a load of 10 N, the friction coefficient of PTFE/PS was about 62% lower than that of pure water, and the wear volume was 86% lower than that of pure water.Increasing the PTFE content can improve the lubricating properties of PTFE/PS. PTFE/PS with 4.0 wt% PTFE addition can reduce the friction coefficient from 0.825 to 0.213, which is nearly 74% lower than that of pure water, and the wear volume of PTFE/PS with 4.0 wt% PTFE is 92.4% lower than that of pure water.The spherical structure of PTFE/PS as a ball bearing transforms the sliding friction into rolling friction, and a low shear transfer film composed of PTFE is formed on the wear surface. The combination of rolling friction and low shear transfer film results in excellent lubrication performance of PTFE/PS.

## Figures and Tables

**Figure 1 materials-16-03071-f001:**
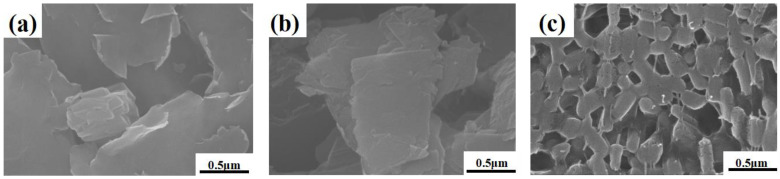
SEM images of (**a**) OMMT, (**b**) EGR, (**c**) PTFE.

**Figure 2 materials-16-03071-f002:**
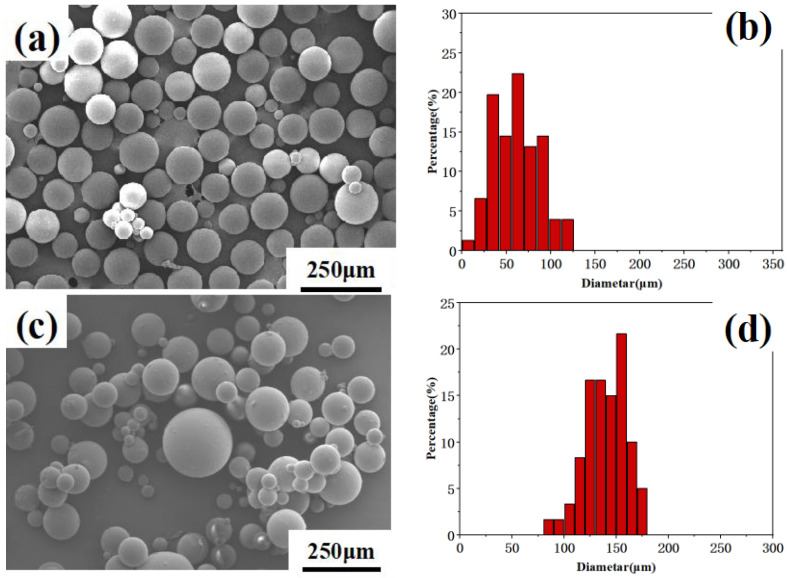
SEM image of (**a**) PS, (**c**) EGR/PS, (**e**) OMMT/EGR/PS and (**g**) PTFE/PS; size distribution of (**b**)PS, (**d**) EGR/PS, (**f**) OMMT/EGR/PS, and (**h**) PTFE/PS.

**Figure 3 materials-16-03071-f003:**
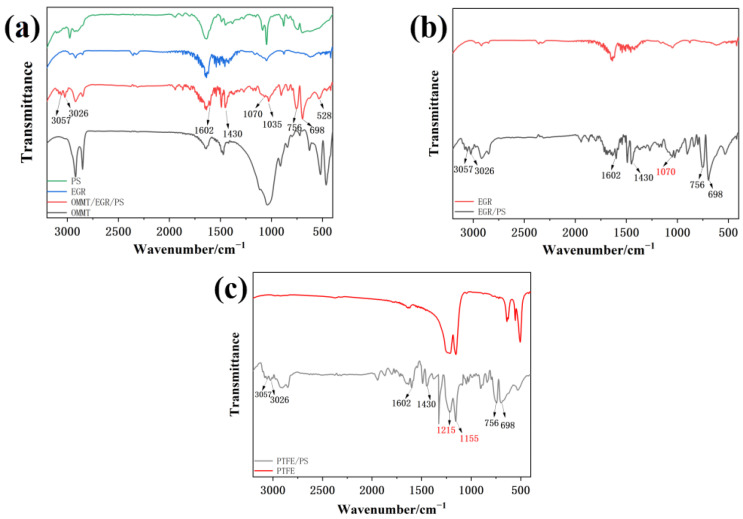
FTIR spectra of (**a**) OMMT/EGR/PS and PS, (**b**) EGR/PS, (**c**) PTFE/PS.

**Figure 4 materials-16-03071-f004:**
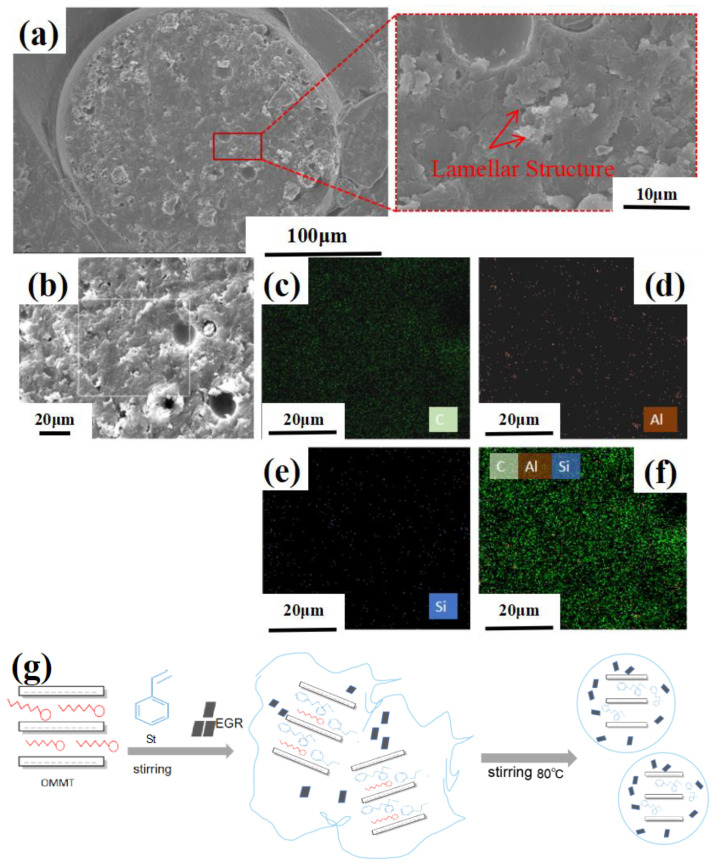
(**a**) SEM image of OMMT/EGR/PS after acetone corrosion and (**b**–**f**) distribution image of its EDS elements; (**g**) schematic of OMMT/EGR/PS polymerization process.

**Figure 5 materials-16-03071-f005:**
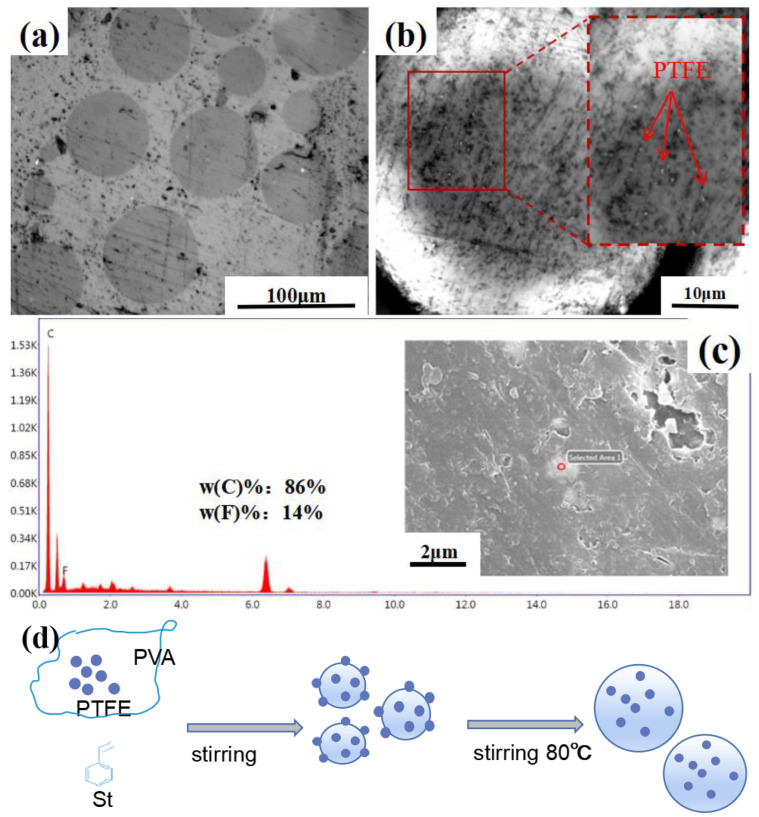
(**a**) Metallograph of PTFE/PS before acetone corrosion, (**b**) metallograph of PTFE/PS after acetone corrosion, (**c**) EDS analysis of small particles in PTFE/PS, (**d**) schematic of PTFE/PS.

**Figure 6 materials-16-03071-f006:**
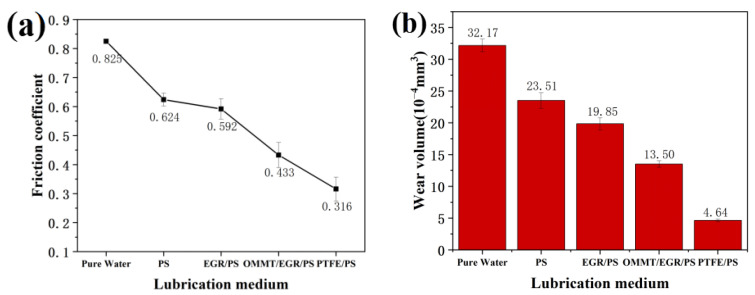
(**a**) The friction coefficient and (**b**) wear volumes of composite microspheres.

**Figure 7 materials-16-03071-f007:**
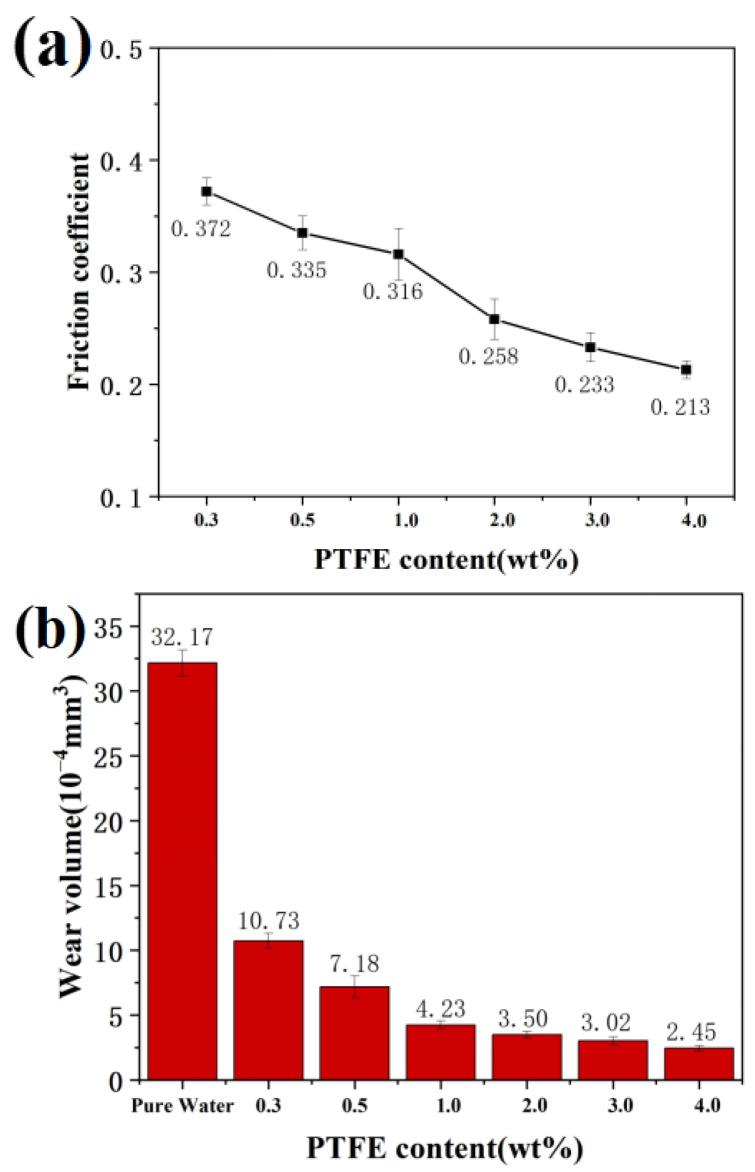
(**a**) The friction coefficient and (**b**) wear volume of PTFE/PS with different content of PTFE.

**Figure 8 materials-16-03071-f008:**
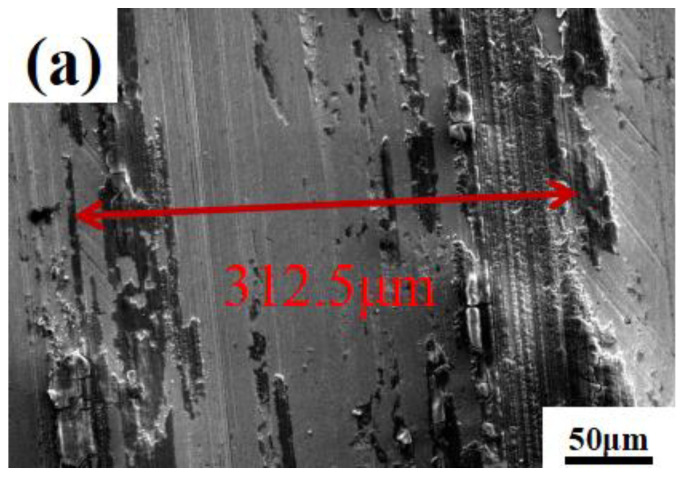
SEM of worn morphology of stainless-steel substrates using (**a**) pure water, (**b**) EGR/PS, (**c**) EGR/PS and (**d**) OMMT/PTFE/PS. 2D scan images of wear tracks (**e**) pure water, (**f**) EGR/PS, (**g**) OMMT/EGR/PS, (**h**) PTFE/PS, (**i**) the roughness of wear tracks.

**Figure 9 materials-16-03071-f009:**
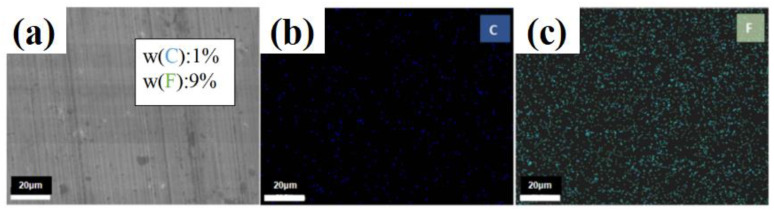
(**a**) EDS analysis of PTFE/PS’s wear track; distribution map of the (**b**) C element and (**c**) F element.

**Figure 10 materials-16-03071-f010:**
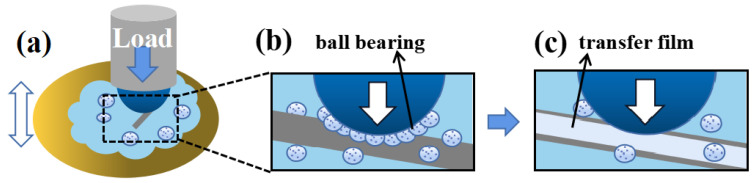
Schematic diagram of (**a**) friction performance test and (**b**,**c**) lubrication mechanism for the PTFE/PS water-based lubricant.

**Table 1 materials-16-03071-t001:** The effect of particle size on the friction coefficient of PTFE/PS.

PTFE Content/wt%	Stirring Speed/rpm	PVA Potency	Average Particle Size/μm	Friction Coefficient
0.3	280	0.5%	74.35	0.412
0.3	380	1.5%	59.85	0.395
0.3	500	2.0%	49.00	0.372
0.5	500	2.0%	47.82	0.335

## Data Availability

The data presented in this study are available in the article.

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
