# Peer review of "Preparation and Lubricating Properties of Polystyrene Composite Microspheres"

_materials, 2023, doi:10.3390/ma16083071_

Round 1

Reviewer 1 Report

The authors describe preparation of polystyrene microsphere composites using different fillers for application in lubricating fluids and show their effect in friction coefficient and wear tests.

The authors apply quite a number of characterization techniques to their materials, though not always in a professional manner (see below for the remarks regarding IR characterization).

One of the motivations referred in the manuscript is a development of lubricating fluids for drilling tools. It might be thus advantageous to demonstrate the efficiency of the developed materials under the real application conditions (in a drilling experiment). The best performance in the friction tests has been found for PTFE-containing PS composites. This is not much surprizing taking into account the excellent lubricating properties of PTFE material. My major concern is that based on the environmental considerations there is a trend to avoid perfluorinated materials as much as possible in future applications. The question arises, whether the developed PS/PTFE composites for lubrication fluids will have reasonable application perspectives and can compete with environmentally benign alternatives. Any analysis of this problem in the Introduction part would be highly welcome.

Further comments on this manuscript can be found below. To sum up, I can recommend publication of this manuscript only after major revisions.

Experimental

Please clarify whether the polymerization was performed in an inert atmosphere, in a closed vessel without degassing, or just in a beaker. It is also unclear how much styrene monomer (g, ml, mol or any other unit) was used per 0.5 g of FG (or other fillers).

What kind of IR-technique was applied? ATR, reflection or transmission configuration? Were any spectral corrections applied?

In general, the description of the friction tests is not sufficiently detailed. What are the amounts of lubricating dispersion used in them? Were the amounts always the same? What was the reciprocation frequency?

Results and discussion

Please clarify how many particles were used in the size distribution analysis for each sample.

In Figure 2a, the same magnification should be used as in c, e and g for a better visualisation and comparability.

Simple IR-analysis cannot provide the information that the fillers “are evenly dispersed” in the polymer, unless IR-mapping is performed (but the resolution will still be in the range of microns in this case). The quality of the spectra is generally very poor. In Fig. 3(a), green line (stated as PS), no signals of PS are seen at all (compare for example with PS-spectrum in SDBS-database). The strongest bands in the spectrum of PTFE/PS shown in Fig. 3(c) are actually due to liquid water in the sample.

Since authors propose the styrene intercalation into OMMT, they should provide a confirmation e.g. by comparison of XRD patterns.

“Figure5(c) gives the SEM image of samples after corrosion.”  I don’t see any SEM image there.

In the wear track profiles shown in Fig. 8, particularly 8(h), the baselines are obviously mismatched. Possibly, the images in Fig. 8(a)-(d) are not correctly scaled. Longer range should be applied for the profile scan to obtain a correct baseline. In this respect, the calculation of the track depth is not precise, deteriorating the discussion of the results.

It has to be kept in mind that the lubricating efficiency may be dependent on the size of the microspheres. Since the particles sizes for different composites are quite different, the results obtained from the friction tests are less unambiguous. It is recommended to perform a series of tests based on the particles of the same composition but different size (prepared with different amounts of added surfactant / different stirring ratio / ultrasonication) or refer to a literature.

The authors are generally encouraged to use more references, but especially when they apply certain subject-related definitions (e.g. transfer film), maybe refer to ISO / ASTM standards in case they were followed.

Ref. 17, journal name is missing

After all, I recommend a comprehensive proofreading of the manuscript to correct grammar errors and to improve the wording.

Reviewer 2 Report

In this manuscript, entitle “Preparation and Lubricating Properties of Polystyrene Compo-2 site Microspheres”, the authors have new lubricant formulations based on suspensions of composite microspheres. The work presents a combination of different raw materials or additives to improve the fundamental properties of such lubricant formulations, like montmorillonite, elastic graphite or PTFE. Although, from my point of view, both the materials section and all the acronyms used throughout the article is hard to follow, the manuscript is well structured with a deep characterization of the synthetized lubricants. However, although this work could be published in Materials, I have some concerns about the results which should be clarified prior its publication;

-          Authors used a silicone nitride ball to carry out the friction tests. This is of great concern to me, since the contact on which they are measuring friction is steel-silicone. Is that close to the actual use of these materials? For bearing applications the tribological contact should be steel-steel.

-          On the other hand, just three repetitions have been performed in such test, are the results a good measure of the reality? This type of tests usually requires between 6 or 7 repetitions because the results are highly variable.

Reviewer 3 Report

This paper aims to assess the lubricating performances of PS composites obtained by suspension polymerization methods. The paper seems well organized with very interesting results that deserve publication. However, I think that authors have to improve the following points before publication:

-           The term “Elastic” graphite (EGR) is probably incorrect. Maybe it would be “Expanded” graphite flakes.

-           A better explanation of the sample preparation is necessary to avoid possible confusion in the reader.

-           Limitations in the suggested approach (i.e. stability, fabrication time, complexity) should be discussed in the conclusions section.

-           The following reference may be of interest to the author and can be added to the revised manuscript:

https://doi.org/10.1002/pc.23444

https://doi.org/10.3390/ma9100856

https://doi.org/10.1016/j.triboint.2016.07.016

https://doi.org/10.1515/polyeng-2020-0302

Round 2

Reviewer 1 Report

After a quick glance at the responses of the authors to my comments as well as at the revised manuscript I would recommend to publish the manuscript with minor corrections, which will be mostly regarding the improvements of the language including the new corrections. For example, the phrase “It is worth noting that some substances in PTFE are difficult to be decomposed…” is not clear: It may mean that only some additives in PTFE have this property, but not PTFE itself. I would recommend to state the following: “It is worth noting that substances like PTFE decompose extremely slow under ambient conditions…”. Another example is the phrase “The deformed vibration of benzene ring skeleton”. The usual term is “deformational vibration”. Unfortunately, there are many such small erroneous wordings in the text so that a proofreading by a specialist would be highly beneficial (check for example the sentence in line 26-27 where the verb is missing).

From the response of the authors to my question about the applied IR technique, I suspect that they performed the sample measurements in KBr pellets. This would explain the poor quality of the spectra, because for a good quality a perfect miscibility of the sample with KBr is a prerequisite. This is why only the spectrum of OMMT is well resolved. The spectrum of PTFE is also okay because of the very small particle size of the material. Typically, polymers do not get easily grinded and mixed with KBr, that is why they are mainly measured using ATR technique. I recommend to consult a specialized literature or an experienced person prior measuring IR spectra next time.
